# Effect of the Bleaching Process on Changes in the Fatty Acid Profile of Raw Hemp Seed Oil (*Cannabis sativa*)

**DOI:** 10.3390/molecules28020769

**Published:** 2023-01-12

**Authors:** Wojciech Golimowski, Mirosława Teleszko, Adam Zając, Dominik Kmiecik, Anna Grygier

**Affiliations:** 1Department of Agroengineering and Quality Analysis, Faculty of Production Engineering, Wroclaw University of Economics and Business, Komandorska 118/120, 53-345 Wrocław, Poland; 2Department of Food Technology and Nutrition, Faculty of Production Engineering, Wroclaw University of Economics and Business, Komandorska 118/120, 53-345 Wrocław, Poland; 3Department of Bioorganic Chemistry, Faculty of Production Engineering, Wroclaw University of Economics and Business, Komandorska 118/120 Street, 53–345 Wroclaw, Poland; 4Department of Food Technology of Plant Origin, Faculty of Food Science and Nutrition, Poznań University of Life Sciences, Wojska Polskiego 31, 60-624 Poznan, Poland

**Keywords:** *Cannabis sativa* L., hemp oil, Finola, Earlina8FC, Secuieni Jubileu, bleaching earth, fatty acid profile

## Abstract

Many refined oils from soybean, rapeseed, and sunflower, among others, are available on the food market, except olive oil. Refining, on the small production scale of niche oils, is not used due to the high cost of the refining process. Unrefined oils are characterized by intense taste, odor, color, and undesirable nutrients. The problem to be solved is determining the effects of incomplete refining of niche oils on their composition. One process, which does not require the use of complex apparatus, is the bleaching process. The results presented in this article relate to the research stage, in which the aim is to evaluate the changes occurring in the oil due to the low-temperature bleaching process with different process parameters. The presented research results provide evidence of the absence of adverse changes in the fatty acid profile of hemp oil of the varieties ‘Finola’, ‘Earlina 8FC’, and ‘Secuieni Jubileu’. Seven different types of bleaching earth were used to bleach the oil in amounts of 2.5 and 5 g/100 g of vegetable oil. The fatty acid profile was obtained by gas chromatography (GC-FID). The obtained chromatograms were subjected to statistical analysis and principal component analysis (PCA). The results show that there was no effect of the type of bleaching earth and its amount on the change in the fatty acid profile of bleached oils. Only real differences between the types of hemp oils were observed. However, an overall positive effect of the bleaching process on hemp oil was found. The amount of saturated fatty acid (SFA) was reduced by 17.1% compared with the initial value, resulting in an increase in the proportion of polyunsaturated fatty acids (PUFA) by 4.4%, resulting in an unsaturated fatty acid (UFA) proportion of 90%. There was a significant improvement in the SFA/PUFA ratio by 26% over the baseline, and the omega-6/omega-3 ratio by 8.9% to a value of 3.1:1. The new knowledge from this study is evidence of the positive effect of the low-temperature bleaching process on the fatty acid profile. In contrast, the parameters of the bleaching process itself are not significant.

## 1. Introduction

Food manufacturers are looking for raw materials and processing methods to bring new foods with unique nutritional and health-promoting properties to the market. In the case of fats, these are unrefined vegetable oils produced from unconventional oilseeds or fruit and vegetable seeds. These are mainly cold-pressed, unrefined oils with individual sensory characteristics that are only sometimes accepted by consumers. Therefore, a study was undertaken to verify the effects of the low-temperature bleaching process (one of the oil refining processes) of hemp oils on their fatty acid profile. The results obtained provide valuable information for the production of niche vegetable oils.

Vegetable oils are raw materials for the production of nutritional and technical materials as well as energy carriers [1]. Soybean, rapeseed, sunflower, and palm oils are produced on an industrial scale [2], pressed or extracted from seeds, and unrefined or refined [3]. Only unrefined pressed oils, made from oilseeds produced on a small scale or made from fruit and vegetable seeds, are available for the small-scale production of niche vegetable oils [4]. They are a source of essential fatty acids (omega-3 and omega-6), phytosterols, phospholipids, carotenoids and many other valuable food components and vitamins [5]. The fatty acid profile is the basic information about a vegetable oil in terms of nutrition [6]. A high proportion of saturated fatty acids (SFAs) is undesirable in the human diet [7,8,9]. In addition to animal fats rich in SFAs, the composition of some vegetable oils is also mainly SFAs. Oils with a high proportion of SFAs, more than 75%, include palm oil and coconut oil; flax, sunflower, canola, hemp, and almond oils contain less than 10% [10]. Some cold-pressed oils are characterized by a predominant percentage of monounsaturated fatty acids (MUFAs). For example, vegetable oils extracted from white mustard or coriander are a valuable source of MUFAs and contain less than 6% SFAs and a large amount of phytosterols [11]. Olive oil and moringa oil also have a high proportion of MUFA, with oleic acid dominating [12]. The presence of oleic acid contributes to lowering the risk of cardiovascular disease as well as cancer and autoimmune diseases [6]. The third fatty acid group in oils includes polyunsaturated fatty acids (PUFAs). PUFAs belong to the group of essential fatty acids that are not produced by our bodies, and we need to supplement them, preferably through a proper diet. Vegetable oils are an important source of these fatty acids. Among PUFAs, we distinguish omega-3 and omega-6 fatty acids. Oils characterized by a high percentage of PUFAs are flaxseed oil [13], evening primrose oil [14], camelina oil [15], hemp oil [16], pumpkin seed oil [17,18,19], black cumin oil [20], grape seed oil [21] and prune seed oil [22]. Each of these oils is characterized by different percentages of omega-6 and omega-3, and studies show that a high ratio of omega-6 acids prevents further conversion of omega-3 acids [23]. Current research seeks to improve the fatty acid composition of cold-pressed oils by preparing oil blends. Research is being carried out in the direction of obtaining oil with the highest possible omega-6/omega-3 ratio, as a source of PUFAs [24]. Garden cress oil with sunflower, rice bran, and sesame, for example, are used for this purpose. The second type of designed blends includes blends of oils in which oxidative transformations are supposed to occur more slowly during frying [25]. For this purpose, rapeseed oil, evening primrose oil, camelina oil, black cumin oil, hemp oil, linseed oil, wheat germ oil, and rice bran oil are used.

Unrefined vegetable oils contain many undesirable components that impart unpleasant taste, odor, and color affecting the functional properties of the oil [3]. Another group of contaminants includes pesticides, aromatic hydrocarbons, and heavy metals in trace amounts, which are determined by the plants’ growing conditions [26]. On the part of both consumers and the food industry, vegetable oils must meet minimum food safety conditions [27]. Oils must be odorless, with a natural lipid taste, colorless, and free of contaminants [3]. Due to the small scale of niche oil production, its complete refining is not possible due to the high cost of the process. The oil refining process is multi-step [28] and is not cost-effective for small volumes of vegetable oil. The refining technology consists of several chemical and physical processes [29]: degumming to eliminate phospholipids and mucilaginous gums [30]; removal of free fatty acids (FFAs), metals and chlorophyll by neutralization [5]; washing and drying to remove trace amounts of water and soaps; bleaching to change color; and deodorization to remove volatiles and carotenoids [31]. Refined oils have a longer shelf life but are devoid of valuable health-promoting pharmaceutical substances [32]. As a result of the refining process, traces of undesirable compounds remain in the oil, such as glyceride esters [33], harmful trans isomers of fatty acids, and polymeric triacylglycerols (TAGs) [34]. As a result of oil refining, sterols, tocopherols and valuable nutrients are removed [34,35]. Kwaśnica et al., in a previous article, presented the lack of a significant effect of the bleaching process on the phytosterol profile of unrefined oils [36]. Niche oils are mechanically extracted oils and are not refined.

Considering the technological aspects, the bleaching process is the least complicated from among the refining processes, but it has some disadvantages. During the bleaching process, there is a loss of up to 2% of the oil by weight [36]. Methods of extracting oil from bleaching earth using solvents are known [37]. Bleaching earths, commonly used in the oil and wine industries, are used for the process [38]. These can include special silicates, adsorption clays, activated carbon, bentonites, and mixtures [39,40,41]. In addition, chemical treatments, most often strong acids, are used to change the textural characteristics of bleaching earth [42,43]. Another method is the chemical or physical treatment of wood to obtain activated carbon with strong adsorptive properties. Bleaching is carried out periodically, bleaching earth is poured into the oil in an amount greater than its adsorptive capacity, in practice up to 5% *m*/*m* of oil, and then the components are mixed for a certain period. The bleaching process is often performed at temperatures above 80 °C [43]. Our research shows the good adsorption properties of the earth also lower oil temperatures [44]. A study by Song at al. presented the positive effect of bleaching fish oil on its fatty acid profile [45].

Qualitative changes occurring in oil products at various refining stages (degumming, neutralization, bleaching, deodorization) have been described primarily in terms of the content of carotenoids, chlorophylls, phytosterols (PS), tocopherols, phospholipids, and free fatty acids. In the case of bleaching, the main technological goal is to improve the color of the oils. However, the fact is that not only color compounds, phosphatide residues, soaps, phospholipid impurities, lipid peroxidation products and other undesirable substances are removed from them at this stage [42,46], but also, in part, phytosterols. Competitive absorption between PS and pigments and the strength of the interactions between the bleaching earth adsorbent and PS are believed to be the reason for the above phenomenon. The use of acid-activated BE in refining edible oils results in the formation of steradians (SD) from sterols via a catalyzed dehydration reaction [47,48]. This problem was discussed in more detail in our previous publication [36]. However, while changes in the content and molecular structure of phytosterols during the production of partially or fully refined oils are fairly well described and recognized, insufficient knowledge exists of the effects of refining on the fatty acid profile of these products. This fact prompted us to take a closer look at this issue.

The purpose of this research was to comprehensively analyze the effect of the bleaching process on the change in hemp oil properties. The characteristics of seeds and hemp oil from the ‘Finola’, ‘Earlina 8FC’ and ‘Secuieni Jubileu’ varieties were presented by Golimowski et al. [16]. Kwaśnica at al. analyzed changes in the phytosterol profile [36]. In the work presented here, the effect of the low-temperature bleaching process on the change in the fatty acid profile of hemp oils was studied.

## 2. Results

The study was concerned with evaluating the significance of the effect of the partial refining process with bleaching earth (BE) on changes in the fatty acid (FA) profile and the values of the health quality indices (LHI) of cold-pressed and heat-pressed hemp oils. The results of the experiment are presented in Table A1, Table A2, Table A3, Table A4, Table A5 and Table A6.

### 2.1. The Results of the MANOVA Multivariate

As can be seen from the MANOVA analysis, the variables studied (type and dose of bleaching earth, pressing temperature, variety) and their interactions significantly (*p* < 0.05) affected the proportion of individual fatty acids in the hemp oils studied, and consequently the changes in the values of lipid quality indices (Table 1 and Table A1, Table A2, Table A3, Table A4, Table A5 and Table A6). At the same time, it was found that the FA profiles and LHI values of unrefined oils and oils purified with BE differ markedly. As a result of bleaching, ΣSFA decreased by 1.07–2.25% relative to the proportion of these FAs in the raw oil. Lignoceric acid (C 24:0) averaged 0.05–0.24% of all FAs in unrefined oils, and peanut acid (C 20:0; average share in crude oils: 0.38–1.17%) was not detected in the samples. The share of stearic acid (C 18:0) decreased from 0.36% (‘S. Jubileu’) to 0.95% (‘Finola’), and behenic acid (C 22:0) from 0.17 to 0.36% (for ‘S. Jubileu’ and ‘Earlina 8FC’ varieties, respectively). In opposition to this trend, ΣUFA in bleached oils was higher than in crude oils by an average of 1.07% (‘S. Jubileu’); 1.84% (‘Earlina 8FC’) and 2.25% (‘Finola’). The main unsaturated acids of hemp oils are linoleic acid (C 18:2 omega-6; LA), linolenic acid (C 18:3 omega-3; ALA), and oleic acid (C 18:1, n-9), which together account for >80% of the total FA profile. As shown in our experiment, there was an increase in the proportion of each of these fatty acids in bleached oils compared with raw oils (Table 1). The changes were most pronounced concerning C 18:2 omega-6 acid, the proportion of which increased by up to 2.8% (average value; variety ‘Finola’). In contrast, the seed oil of ‘S. Jubileu’ after the bleaching process marked the greatest increase in the proportion of C 18:3 omega-3 acid (average 1.86%).

When considering the effect of BE type on the FA profile of bleached oils, it was noted that the use of chemically modified kerolite-hydrated magnesium silicate of pH 6 (P7) proved most favorable for hemp oils from the ‘Earlina 8FC’ and ‘S. Jubileu’ varieties. Samples bleached in this way had the highest average proportion of UFA (90.58 and 89.85%, respectively) and, simultaneously, the lowest proportion of SFA (9.42 and 10.15%, respectively). This is an interesting observation given the results of our previous experiment, in which the effects of the same bleaching earth on the phytosterol profile of hemp oil were analyzed. Indeed, diatomaceous earth P7 showed the strongest adsorption properties towards phytosterols [36].

Changes in the FA profile observed in bleached hemp oils affected the LHI values. The results of this analysis are included in Table 1. One of the most important parameters for estimating the health-promoting potential of edible oils is the PUFA/SFA ratio. According to Chen and Liu [49], the PUFA/SFA ratio is used to assess the effect of diet on cardiovascular function. It is generally accepted that SFAs are hypercholesterolemic, while PUFAs lower serum cholesterol levels. Hence, oils with higher PUFA/SFA values are considered more beneficial to health. Increasing the proportion of UFAs while reducing the proportion of SFAs in bleached oils contributed to a marked increase in the value of the index in question relative to crude oils. Its highest average value was determined in bleached oils of the ‘Earlina’ variety (8.24, which means an increase in this LHI by 1.71 compared with unbleached oil). From a nutritional point of view, an important quality characteristic of edible fats is the reciprocal ratio of omega-6/omega-3 acids. As shown in earlier studies of our own [16], hemp oils are characterized by a very good value for this discriminant (about 3:1). The bleaching process did not significantly affect the relationship between the proportions of FAs of the omega-6 and omega-3 family; however, comparing bleached oils with crude oils in this regard, a slight decrease in the value of the aforementioned LHI can be observed (a favorable phenomenon). A similar direction of change was also noted for the other lipid quality indices, i.e., AI (atherogenicity index), h/H (hypocholesterolemic/hypercholesterolemic index), and TI (thrombogenicity index) (Table A1, Table A2, Table A3, Table A4, Table A5 and Table A6).

It is generally believed that the refining process does not cause significant changes in the fatty acid profile of oils [50,51,52,53], hence this issue is rarely addressed in the literature. Nevertheless, Song et al. [45], who investigated the effect of chemical refining, including bleaching, on the FA profile in fish oil, found that the profile changed significantly at different stages of the process. The sums of SFA in the crude, de-sluiced, deacidified, bleached, and deodorized oil were 43.89%, 37.02%, 35.10%, 34.80%, and 33.92%, respectively. This confirms our observation that SFAs are reduced during refining. As the authors pointed out, EPA and DHA acids in the total PUFA pool had the highest proportion regardless of the refining stage. In contrast, the relative contents of EPA and DHA steadily increased during the process [45]. An analogous situation was observed for LA and ALA acids in our study.

### 2.2. Principal Component Analysis (PCA)

To observe possible clusters of fatty acid composition in fresh and bleached hemp oil the principal component analysis (PCA) method was applied. The result of the distribution of the analyzed samples depending on the differentiating factor is shown in Figure 1. The first two principal factors accounted for 76.2% (Dim1 = 50.3% and Dim2 = 25.9%) of the total variation. Factor 1 was mainly correlated with the share of C 18:0 (r = 0.922), C 18:1 (r = 0.900), and MUFA (r = 0.901) and negatively correlated with the share of C 18:2 omega-6 (r = −0.934) and PUFA (r = −0.991). Factor 2 was mainly correlated with the share of C 16:0 (r = 0.883).

The data shown in the score plots divides the analyzed samples into six groups. The clustering is based on the seed variety used during the process and the temperature at which the process was conducted. Above the X axis are samples obtained from seeds of the ‘Finola’ variety. The oils obtained from the ‘Finola’ variety also have the greatest variation. Below the X-axis are the oils obtained from the ‘Earlina’ variety (left of the Y-axis) and ‘S. Jubileu’ variety (right of the Y-axis). The oil obtained from the ‘Finola’ and ‘Earlina’ varieties had similar proportions of PUFA and MUFA. However, ‘Finola’, compared with ‘Earlina’, was characterized by a higher share of C 16:8 and a lower proportion of C 18:0. The oil obtained from the ‘S. Jubileu’ variety was distinguished from the others by a higher share of MUFA and a lower share of PUFA.

The temperature of the pressing process was also of great importance for the clustering of the sample. Pressing at an elevated temperature led to less dispersion of the analyzed samples. When pressing was carried out at a low temperature, the samples were characterized by high variation. The clustering of samples was not affected by the bleaching process conditions. Neither varying the dose of earth nor its type was significant.

PCA was also used to evaluate the change in fatty acid composition for pressed oil and bleached oils (Figure 2). The first two principal factors accounted for 81.2% (Dim1 = 57.5% and Dim2 = 23.7%) of the total variation.

Samples of fresh, unbleached oils form a new group located at the extreme right of the Y axis. As before, samples of oils obtained from the ‘Finola’ variety are located above the X-axis, and other pressed oils are located below the X-axis. The oils obtained from the ‘Earlina’ and ‘S.Jubileu’ varieties are located close to each other. The outlier samples are the oils obtained from the ‘Finola’ variety.

Unbleached oils are characterized by a higher share of SFA and MUFA and a lower share of PUFAs compared with samples purified by bleaching with seven different bleaching earths.

Based on the collected data, it can be concluded that the bleaching process positively changes the fatty acid profile. Figure 3 shows how the bleaching process affects the summary proportions of SFA and PUFA and the summary proportion of UFA unsaturated acids. It can be seen that there are significant differences in the grouped results, independent of the process parameters and type of oil. In the case of the average values of the SFA share, a group of acids not recommended in cooking oils, their share was reduced by 17.1% relative to the base value. As a result, the share of polyunsaturated acids (PUFA) increased by 4.4%, increasing the share of saturated acids UFA in oils by 2.3%.

The second important factor in the fatty acid profile is the PUFA to SFA ratio. The higher the ratio, the lighter the oil is for consumption. As a result of the bleaching process, a significant change in this ratio of 26% was noted relative to the baseline variance (Figure 4). The ratio of n-6 to n-3 fatty acids in terms of diet should be 2:1. In the case of hemp oils, it is on average 3.4:1. As a result of the bleaching process, this ratio was found to be reduced by an average of 8.6% to 3.1:1 (Figure 4).

### 2.3. Summary of Results Analysis

Based on statistical analysis, it can be concluded that all free parameters had a significant effect on changing the fatty acid profile (Table 1). Important, from the point of view of human nutrition, was the level of reduction in saturated acids (Figure 3 and Figure 4). Using bleaching earth and a low bleaching temperature, there is greater adsorption of saturated acids relative to non-saturated acids, which may be due to the porous structure of the bed. This hypothesis is confirmed by PCA analysis, the result of which is that, despite the significant influence, none of the paramaters showed a strong correlation (Figure 1 and Figure 2).

## 3. Materials and Methods

Hemp seeds harvested in 2021 in Poland were used for the study. Three varieties of cannabis were used: ‘Finola’, well known in the literature, and the poorly recognized varieties ‘Ealina 8 FC’ and ‘Seciueni Jubileu’. Seeds were dried by the seed producer and immediately delivered to Wroclaw. Fresh, unseasoned seeds were pressed. Analyses and the oil blanching process were performed at the laboratory of the Wroclaw University of Economics and Business and the University of Life Sciences in Poznan. The oils obtained for the study were stored in a cool and sunlight-free place. The bleaching process was performed on bleaching soils supplied by manufacturers in Europe.

### 3.1. Materials

Raw hemp oil from the seed varieties ‘Earlina 8FC’, ‘Finola’ and ‘Scuieni Jubileu’ was used for the study. The oils were pressed using the one-step unheated (20 °C) method and heated (60 °C) seeds. Published characteristics of the seeds, oil, and pressing parameters are described by Golimowski et al. [16]. In order to better compare the results, the fatty acid profiles of the obtained vegetable oils are presented in Table 2.

Seven different bleaching earths, commercially available and dedicated to refining cooking oil, were used for the study, their characteristics described in detail by Kwaśnica et al. [36]. Used were:-Attapulgite clay: physically activated (P1), modified with pH 3.2 (P2), and unmodified (P3). This consists mainly of SiO_2_ (55–60%) and has a high proportion of Fe_2_O_3_ (12–14%).-Magnesium bentonite: unmodified with pH 8.5 (P4) and pH 10 (P5) and acid activated (P6). This consists mainly of SiO_2_ (59–63%) and has a high proportion of MgO (23%).-Kerolite-hydrated magnesium silicate, chemically modified with PH 6 (P7). This consists mainly of SiO_2_ (53%) and is characterized by a high proportion of MgO (30.5%).

### 3.2. Methods

Bleaching was performed on a laboratory scale. A total of 100 ± 0.01 g of oil was taken. The oil was stirred with simultaneous heating to 50–60 °C. After the minimum temperature was reached and stabilized, 2.5 ± 0.05 g or 5 ± 0.05 g of bleaching earth was poured in. The mixture was stirred for 10 min at a constant magnetic stirrer speed of 550 rpm (Figure 5).

After the bleaching process, the mixture was poured into a Buchner funnel (Figure 6). Cellulose filters were used for filtration, on which bleaching earth was deposited. Filtration continued until a dry post-filter cake was obtained. The filtration time was unique and depended on the amount and type of soil used.

The fatty acid methyl esters (FAMEs) were quantified by a gas chromatography method using a J & W Scientific HP-88 series 100 m × 0.25 mm × 0.20 μm fused silica capillary column (Agilent Tech. Inc., St. Clara, CA, USA) and flame-ionization detector (FID) from Agilent Tech. Hemp oils were saponified with 0.5 M KOH in methanol. Transesterification of fatty acids with the BF_3_ (boron trifluoride) solution in methanol was carried out using the official American Oil Chemists’ Society (AOCS) Ce 2–66 method [54]. We used a 7890 A series gas chromatograph (Agilent Tech. Inc., St. Clara, CA, USA) at an injection volume of 1.0 mL and a split ratio of 1/50. Helium was used as the carrier gas at a head pressure of 2.0 mL/min at a constant flow. Air, hydrogen, and helium make-up gas flow rates by FID (Agilent Tech. Inc., St. Clara, CA, USA) were 450, 40, and 30 mL/min, respectively. The detector and injector temperatures were chosen as 280 °C and 250 °C, respectively. The initial column temperature of 120 °C was held for 1 min, increased to 175 °C at 10 °C/min and then held for 10 min. It was then increased to 210 °C at 5 °C/min, held for 5 min, increased to 230 °C at a rate of 5 °C/min, and maintained for 5 min. The conditions for chromatographic separation conformed with the procedure described by Wołoszyn et al. [55].

According to the fatty acid profile analysis, selected nutritional quality parameters of the hemp oils were calculated:PUFA/SFA ratio = (ΣDiUFA + ΣTriUFA + ΣTetraUFA)/ΣSFA [49];n-6/n-3 PUFA ratio = (C18:2 n-6 + C18:3 n-6)/(C18:3 n-3 + C18:4 n-3) [49];AI (atherogenicity index) = (C12:0 + 4 × C14:0 + C16:0)/ΣUFA [49];TI (thrombogenicity index) = (C14:0 + C16:0 + C18:0)/[(0.5 × MUFA) + (0.5 × Σn-6) + (3 × Σn-3) + (Σn-3/Σn-6)] [56];h/H (hypocholesterolemic/hypercholesterolemic Index) = [(C18:1 n-9 + C18:1 n-7 + C18:2 n-6 + C18:3 n-6 + C18:3 n-3 + C20:3 n-6 + C20:4 n-6 + C20:5 n-3 + C22:4 n-6 + C22:5 n-3 + C22:6 n-3)/(C14:0 + C16:0)] [57].

Two methods, MANOVA and PCA statistical analysis, were used to analyze the results. PCA analysis was performed using the program RStudio (version 2022.07.01 + 554 with the packages FactoMineR v.2.4 and factoextra v.1.0.7). Statistical analysis was performed using MANOVA (multivariate analysis of variance; multivariate Wilks test; Statistica 13.3 (StatSoft, Cracow, Poland). A *p*-value < 0.05 was considered to indicate a statistically significant difference. Chromatographic data (fatty acid analysis) and LHI values are presented as the mean ± SD (*n* = 3)

## 4. Conclusions

The process of low-temperature bleaching of oil has a significant effect on changing the fatty acid profile. Based on statistical analysis, it can be concluded that all factors, i.e., seed temperature, type of oil, amount of bleaching earth and its type, have a significant effect on the fatty acid profile. Based on PCA analysis, it can be concluded that the fatty acid profile differs only between the types of oil and between bleached and unbleached oil. The soil type, its amount, and the pressed seeds’ temperatures were not significant factors. In practice, the differences observed statistically are not significant. The fatty acid profile of hemp oil, independent of the process parameters, changed favorably. An overall positive effect of the bleaching process on hemp oil was found. The amount of SFA was reduced by 17.1% relative to the base value, which increased the proportion of PUFA by 4.4%, resulting in a proportion of UFA of 90%. There was a significant improvement in the SFA/PUFA ratio by 26% relative to the base value and the omega-6/omega-3 ratio by 8.9% to a level of 3.1:1. On this basis, it can be concluded that the low bleaching temperature (60 °C) results in the capture of saturated acids whose freezing point is above 50 °C. The porous structure of the bed caused the retention of saturated acids, resulting in an improvement in the SFA/PUFA ratio. It can be assumed that lowering the temperature can have a more desirable effect, but this is technologically problematic when filtering high-viscosity oil.

Based on the study’s results, there was a positive effect on the fatty acid profile. The lack of differences in the effectiveness of BE allows their unlimited use to improve other oil parameters, with the guarantee of no deterioration of the fatty acid profile. Low-temperature bleaching technology can be recommended for the refining of niche edible oil.

## Figures and Tables

**Figure 1 molecules-28-00769-f001:**
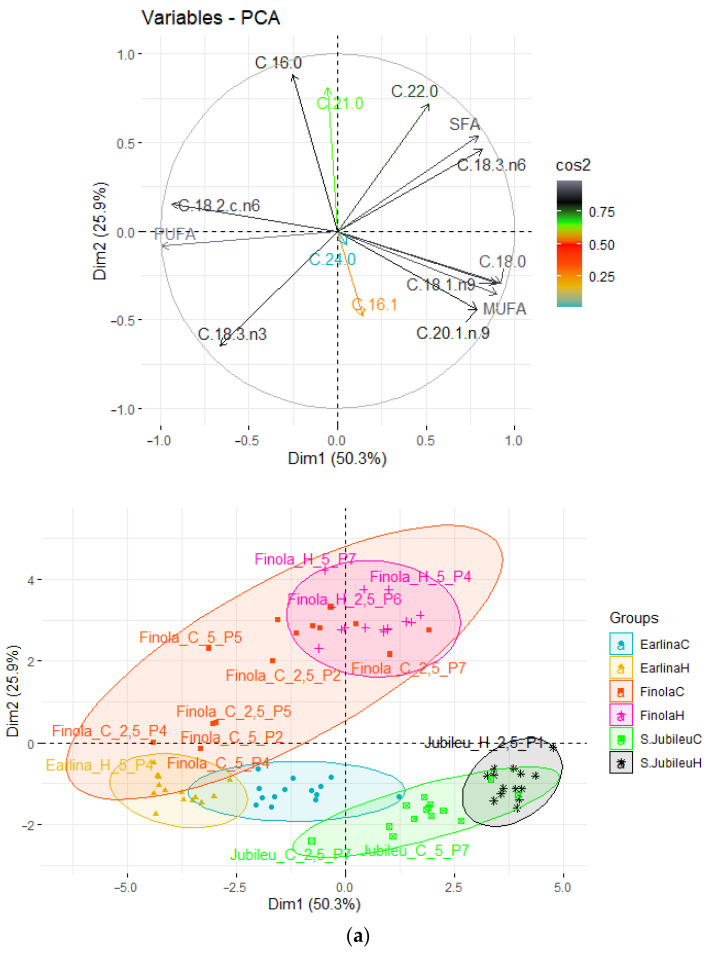
Principal component analysis (PCA) of the loading plot and score plots of data from the fatty acid profiles of bleached hemp oils, where: (**a**) grouped oil type; (**b**) grouped amounts of BE; (**c**) grouped temperature of pressed seeds.

**Figure 2 molecules-28-00769-f002:**
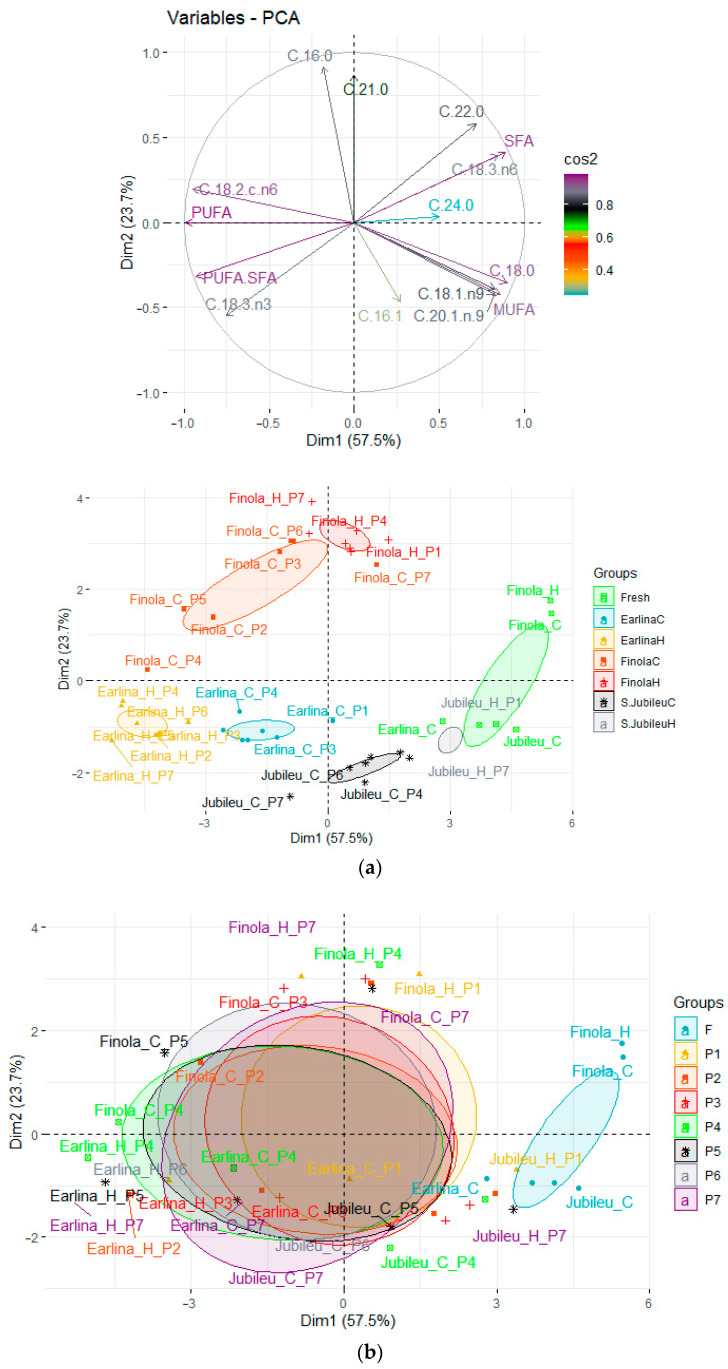
Principal component analysis (PCA) of the loading plot and score plots of data from the fatty acid profiles of fresh and bleached hemp oils, where: (**a**) grouped oil type; (**b**) grouped BE type.

**Figure 3 molecules-28-00769-f003:**
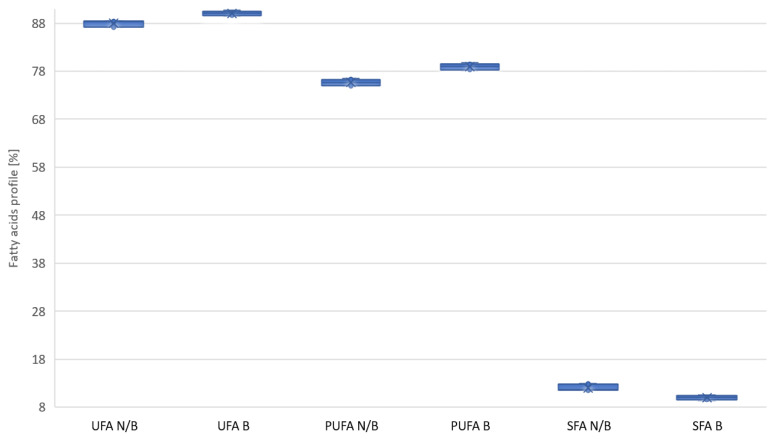
Change in the proportions of SFA, PUFA, and UFA acids after the oil bleaching process. N/B—unbleached oil; B—bleached oil.

**Figure 4 molecules-28-00769-f004:**
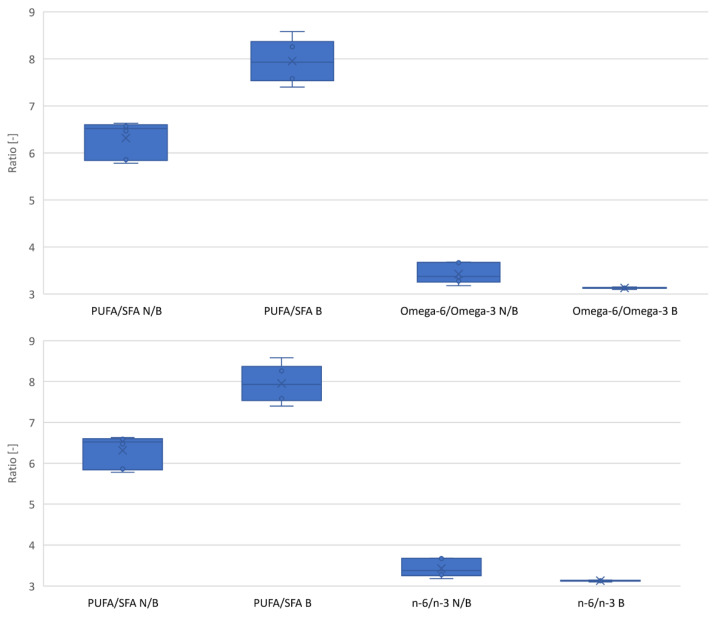
Change in the SFA/PUFA and omega-6/omega-3 ratios due to the bleaching process. N/B—unbleached oil; B—bleached oil.

**Figure 5 molecules-28-00769-f005:**
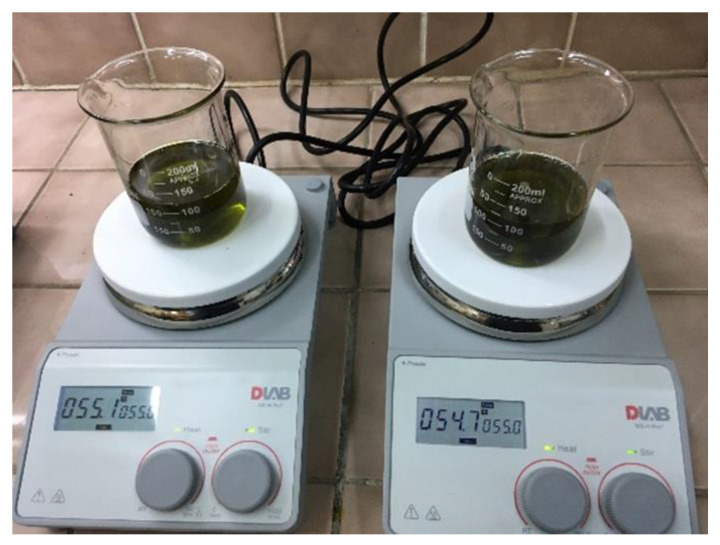
Preparation of hemp oil for bleaching.

**Figure 6 molecules-28-00769-f006:**
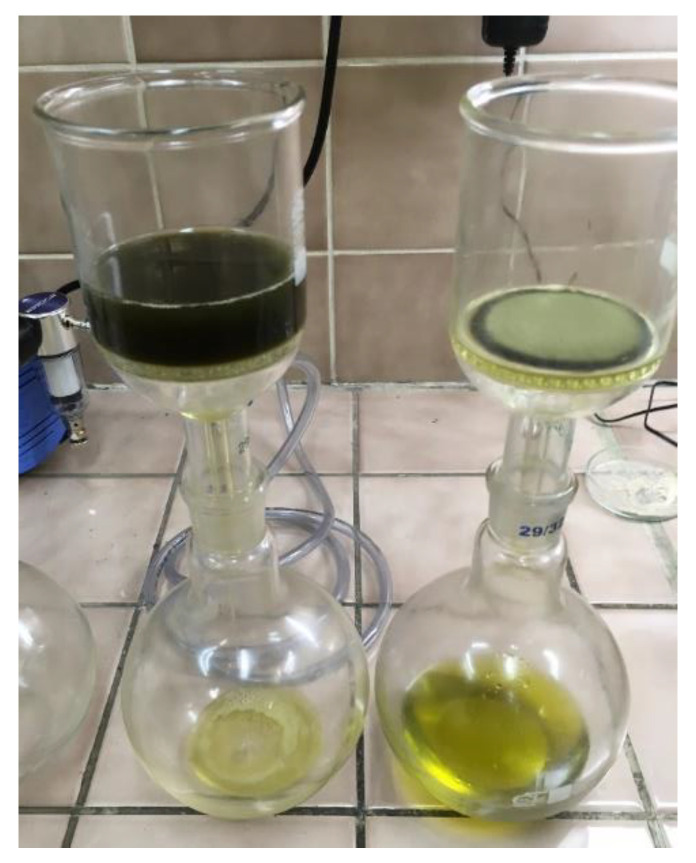
Separation of bleaching earth from oil.

**Table 1 molecules-28-00769-t001:** The results of the MANOVA multivariate test for fatty acids analysis and lipid quality indicators (Wilks test; *p* < 0.05).

Effect	Value	F	df Effect	df Error	*p*
Fatty acid analysis					
Free parameter	0.000000	233,598,416	10	75.0000	0.000000
Variety (1)	0.000029	1378	20	150.0000	0.000000
Temperature (2)	0.085141	81	10	75.0000	0.000000
Bleaching earth (3)	0.004518	12	60	398.0039	0.000000
Dose of bleaching earth (4)	0.390720	12	10	75.0000	0.000000
Variety × Temperature	0.004903	100	20	150.0000	0.000000
Variety × Bleaching earth	0.000642	8	120	596.8110	0.000000
Temperature × Bleaching earth	0.014483	8	60	398.0039	0.000000
Variety × Dose of bleaching earth	0.578527	2	20	150.0000	0.001797
Temperature × Dose of bleaching earth	0.702578	3	10	75.0000	0.001924
Bleaching earth × Dose of bleaching earth	0.059800	5	60	398.0039	0.000000
1 × 2 × 3	0.000782	8	120	596.8110	0.000000
1 × 2 × 4	0.597369	2	20	150.0000	0.003856
1 × 3 × 4	0.008054	4	120	596.8110	0.000000
2 × 3 × 4	0.032313	6	60	398.0039	0.000000
1 × 2 × 3 × 4	0.012914	4	120	596.8110	0.000000
Lipid quality indicators					
Free parameter	0.000000	309,395,916,788	8	77.000	0.00000
Variety (1)	0.000077	1086.13	16	154.000	0.00000
Temperature (2)	0.130364	64.21	8	77.000	0.00000
Bleaching earth (3)	0.017489	10.18	48	382.935	0.00000
Dose of bleaching earth (4)	0.417654	13.42	8	77.000	0.00000
Variety × Temperature	0.015616	67.40	16	154.000	0.00000
Variety × Bleaching earth	0.002345	8.03	96	528.964	0.00000
Temperature × Bleaching earth	0.026803	8.67	48	382.935	0.00000
Variety × Dose of bleaching earth	0.651385	2.30	16	154.000	0.00470
Temperature × Dose of bleaching earth	0.882901	1.28	8	77.000	0.26816
Bleaching earth × Dose of bleaching earth	0.064821	5.93	48	382.935	0.00000
1 × 2 × 3	0.001566	8.86	96	528.964	0.00000
1 × 2 × 4	0.719358	1.72	16	154.000	0.04751
1 × 3 × 4	0.021579	4.23	96	528.964	0.00000
2 × 3 × 4	0.046746	6.89	16	382.935	0.00000
1 × 2 × 3 × 4	0.033623	3.61	96	528.964	0.00000

**Table 2 molecules-28-00769-t002:** Fatty acid profiles [%] and lipid quality indicators in oils obtained from different hemp varieties [16].

Hemp Seeds Oils						
	‘Finola’ (C)	‘Finola’ (H)	‘Earlina’ (C)	‘Earlina’ (H)	‘S. Jubileu’ (C)	‘S. Jubileu’ (H)
C16:0	6.52 ± 0.01	6.61 ± 0.02	6.25 ± 0.01	6.30 ± 0.01	6.27 ± 0.00	6.28 ± 0.02
C16:1 n-7	0.07 ± 0.01	0.07 ± 0.01	0.08 ± 0.00	0.09 ± 0.01	0.06 ± 0.02	0.05 ± 0.00
C17:0	0.02 ± 0.00	0.01 ± 0.00	nd	nd	nd	nd
C18:0	3.20 ± 0.01	3.19 ± 0.01	3.10 ± 0.02	3.21 ± 0.02	3.53 ± 0.03	3.60 ± 0.05
C18:1 n-9	10.67 ± 0.01	10.56 ± 0.05	10.34 ± 0.00	10.82 ± 0.01	11.49 ± 0.13	11.62 ± 0.00
C18:1 n-7	0.93 ± 0.00	1.03 ± 0.05	1.02 ± 0.01	0.77 ± 0.16	0.89 ± 0.18	0.67 ± 0.06
C18:2 n-6	54.70 ± 0.01	54.66 ± 0.04	54.86 ± 0.07	54.86 ± 0.04	54.56 ± 0.09	54.67 ± 0.15
C18:3 n-6	4.054 ± 0.02	4.25 ± 0.08	3.25 ± 0.05	3.61 ± 0.01	3.62 ± 0.12	3.77 ± 0.23
C18:3 n-3	16.02 ± 0.01	16.03 ± 0.01	18.29 ± 0.08	17.81 ± 0.01	17.25 ± 0.01	17.26 ± 0.04
C20:0	1.26 ± 0.00	1.07 ± 0.09	0.70 ± 0.03	0.43 ± 0.00	0.45 ± 0.02	0.34 ± 0.13
C20:1 n-9	0.54 ± 0.00	0.52 ± 0.01	0.47 ± 0.00	0.49 ± 0.00	0.46 ± 0.01	0.48 ± 0.02
C20:2 n-9	0.07 ± 0.00	0.07 ± 0.01	0.05 ± 0.00	0.05 ± 0.01	nd	nd
C21:0	1.14 ± 0.00	1.13 ± 0.01	1.00 ± 0.00	0.93 ± 0.03	0.90 ± 0.00	0.90 ± 0.00
C22:0	0.56 ± 0.03	0.57± 0.01	0.40 ± 0.05	0.42 ± 0.01	0.40 ± 0.05	0.37 ± 0.04
C24:0	0.25 ± 0.02	0.23 ± 0.03	0.19 ± 0.01	0.21 ± 0.00	0.11 ± 0.07	0.00 ± 0.00
SFA [%]	12.95 ± 0.04	12.81 ± 0.10	11.64 ± 0.11	11.50 ± 0.06	11.66 ± 0.01	11.48 ± 0.15
UFA [%]	87.05± 0.06	87.19 ± 0.11	88.36 ± 0.11	88.50 ± 0.12	88.34 ± 0.15	88.52 ± 0.08
MUFA [%]	12.21 ± 0.01	12.18 ± 0.02	11.91 ± 0.01	12.17 ± 0.16	12.89 ± 0.07	12.82 ± 0.04
PUFA [%]	74.84 ± 0.04	75.01 ± 0.13	76.45 ± 0.10	76.26 ± 0.05	75.44 ± 0.22	75.70 ± 0.04
PUFA/SFA ratio	5.78 ± 0.02	5.86 ± 0.05	6.57 ± 0.07	6.63 ± 0.03	6.47 ± 0.02	6.59 ± 0.09
n-6/n-3 ratio	3.67 ± 0.00	3.68 ± 0.01	3.18 ± 0.01	3.28 ± 0.00	3.37 ± 0.01	3.38 ± 0.01
AI	0.07 ± 0.00	0.08 ± 0.00	0.07 ± 0.00	0.07 ± 0.00	0.07 ± 0.00	0.07 ± 0.00
h/H	13.25 ± 0.00	13.09 ± 0.05	14.04 ± 0.01	13.94 ± 0.03	14.00 ± 0.03	14.01 ± 0.06
TI	0.12 ± 0.00	0.12 ± 0.00	0.10 ± 0.00	0.11 ± 0.00	0.11 ± 0.00	0.11 ± 0.00

nd—not detected; C—cold-pressed seeds; H—hot-pressed seeds; AI—atherogenicity index; h/H—hypocholesterolemic/hypercholesterolemic index; TI—thrombogenicity index.

## Data Availability

Not applicable.

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
