# Peer review of "Effect of the Bleaching Process on Changes in the Fatty Acid Profile of Raw Hemp Seed Oil (*Cannabis sativa*)"

_molecules, 2023, doi:10.3390/molecules28020769_

Round 1
Author Response
Thank you for your honest review. When planning the study, we expected to obtain results showing that the soil had no effect on the fatty acid profile. Out of 84 samples, we obtained reproducible results that prove coexistent changes in the fatty acid profile due to the bleaching process. We made the decision to publish the results in a separate article.
Responses to comments:
- I would suggest as title: Effect of bleaching process on changes in fatty acid profile of raw hemp seed oil (Cannabis sativa L)
I agree, the title should be clarified.
- Language should be revised (many grammatical mistakes and punctuation errors in the text
Once again, the article was analysed for language. A proofreading of the text was performed.
- The Abstract must be rewritten to be representative of the whole paper (background and objectives, methodology, main results).
Rewrote entire abstract to indicate relevant information consistent with article content.
- Please provide clear statement on the utility and novelty of this work in abstract.
New is the provision of evidence of the positive effect of the low-temperature bleaching process on changing the fatty acid profile. From the literature, the bleaching process does not affect the fatty acid profile.
- I think. The percentage (%) is not an SI unit. I suggest changing percentage (%) to g/ 100g or ml/ 100ml.
It has been changed
- Line 23 : please add the detector GC-FID 2
It has been changed
- Line 24: What does PCA means
Multicriterial analysis method of research results: principal component analysis (PCA)
- The introduction does not discuss the economic importance of the study
In the first paragraph of the introduction, a pledge of research results for the food industry was added.
- The originality of the study is not focused in introduction section
The originality of the study stems from providing evidence of the effect of different bleaching earths and process parameters (low bleaching temperature) on the fatty acid profile. The literature found a lack of correlation due to the use of bleaching earths at process temperatures >90C. Based on the results of the study, it was proved that the bleaching process has a positive effect on the change in the fatty acid profile regardless of the vintage and amount of earth. This is new information highlighted in the abstract and introduction.
- Please change (n-3 and n-6), by (Omega-3 and Omega-6 )
- Figures should be represented in higher resolution
- The introduction should be enriched with recent references (2017-2022) for example :
- Line 46: Acids SFAs in the human diet is undesirable (Please add a new reference)
Was added:
- Oteng, A.-B.; Kersten, S. Mechanisms of Action of trans Fatty Acids. Adv. Nutr. 2020, 11, 697–708.
- Unger; Torres-Gonzalez; Kraft Dairy Fat Consumption and the Risk of Metabolic Syndrome: An Examination of the Saturated Fatty Acids in Dairy. Nutrients 2019, 11, 2200.
- Merra, G.; Noce, A.; Marrone, G.; Cintoni, M.; Tarsitano, M.G.; Capacci, A.; De Lorenzo, A. Influence of Mediterranean Diet on Human Gut Microbiota. Nutrients 2020, 13, 7.
- Line 49 : predominant percentage of Monounsaturated Fatty Acids MUFAs (Please add a new reference).
“Some cold-pressed oils are characterized by a predominant percentage of Monounsaturated Fatty Acids MUFAs. For example, vegetable oils extracted from white mustard or coriander are a valuable source of MUFAs and contain less than 6% SFAs and a large amount of phytosterols [11].”
11 Kozłowska, M.; Gruczyńska, E.; Ścibisz, I.; Rudzińska, M. Fatty acids and sterols composition, and antioxidant activity of oils extracted from plant seeds. Food Chem. 2016, 213, 450–456.
- What specific bleaching earth was used? in what quantity?
The characteristics of bleaching earths are described in the Materials and Methods chapter. A more detailed characteristics is in the publication [32].
Kwaśnica, A.; Marcinkowski, D.; Kmiecik, D.; Grygier, A.; Golimowski, W. Analysis of Changes in the Amount of Phytosterols after the Bleaching Process of Hemp Oils. Molecules 2022, 27, 7196. doi: https://doi.org/10.3390/ molecules27217196.
The amount of herb also is described in the publication as 2.5g and 5g per 100g of oil.
- Line 72 : The refining technology consists of several chemical and physical processes. I think there's also enzymatic refining. Please see this paper https://doi.org/10.1155/2022/6627013.
Thank you for submitting a very interesting literature item. added to justify the description.
- Refined oils have a longer shelf life. I think, It’s not correct One of the main disadvantages of he refining technology is the loss of substances responsible for healthy, pharmaceutical properties and technological interest in the oils, such as tocopherols, , polyphenols, phytosterols …..and this directly influence decrease in the shelf life of oils and the nutritional quality
I agree. However, in the context of refined oil vs. unrefined oil, refined oils have a longer shelf life. First of all, this is due to the acid number and free fatty acids that are removed by refining from raw oil. In a publication by Kwaśnica et al. we presented the effect of bleaching on changing the profile of phytosterols in the oil. The bleaching process does not cause significant changes, which makes it a convenient process for incomplete oil refining. The last in our series of publications will deal with chlorophyll, carotenoids, and oil color after bleaching. I hope it will be published in the spring.
- Why the authors did not use other analyses that determine the quality and the oxidation state of the oils like: acidity, peroxide value para anisidine value, Trans fatty acids, sterols and Tocopherols
A number of analyses have been carried out, i.e. the effect of the bleaching process on: sterol profile (already published), fatty acid profile (in progress), acid number, level of reduction of carotenoids and chlorophylls, evaluation of color change (in preparation for publication), change of P,Ca,Mg elemental content (in progress of analysis). Spectroscopic analysis of oil structure change (data under analysis). The database is very large. We have adopted a strategy to spread the research results over several articles in order to present the research results in detail.
- Results are expressed as the relative percentage of area of each individual fatty acid peak », It will be better to do analyse with internal standard, and quantify the result.
During the preparation of the trials, we discussed this. The aim of our research was to identify only the changes occurring in oils due to the bleaching process. We expected to find no significant differences, and relied on information from the literature. Repeating the tests on the collected samples may not give measurable results due to their rather long storage time. We are considering using internal standards in future studies.
Reviewer 2 Report
The content of the work is well presented but my only concern is whether the format of the journal was well followed. Why present results before material and methods?
Author Response
Thank you for reviewing the article. The form of the results presented, where the discussion of the results of the study is through materials and methods, is due to the template MDPI.
Reviewer 3 Report
The authors have done extensive work. However, there is plenty of room for improvement. Some issues must be resolved because the results and discussion are not well presented.

Author Response
General
- i) Spell out the full term of all acronyms at its first mention, indicate its abbreviation in parenthesis and please use the abbreviation from then on. E.g Line 24 – PCA, Line 27 – SFA, Line 28 – PUFA, Line 29 – UFA, Line 46 - saturated fatty acids (SFAs), Line 268 – FAME, Line 271 - AOC
Corrected
- ii) Use same theme font for the affiliation (line 11)
Corrected
iii) Some sentences are grammatically correct but the meaning is not clear. Please check throughout the manuscript.
Checked
Introduction
- i) Line 38 – ‘..technical?..”
Corrected
- ii) Line 47 – (SFAs>75%). Please put the reference(s)
Reference are of the end of next sentence [7]
iii) Line 52 – “..cancer, autoimmune, inflammatory diseases.”
Corrected
- iv) “on the one hand”. Please use synonym
Corrected
- v) Line 62 – Sunflower, rice, sesame are not proper nouns. Please check
Corrected
- vi) Line 70 – 71 – “In the case … is impossible” Please rephrase. vii) Line 74 – Washing, Bleaching are not a proper nouns
Corrected
viii) Line 81 – Considering technological considerations (please use synonym).
Corrected
Result & Discussion
- i) Line 100 – 111 should be in the introduction.
We agree with this comment, the text has been moved to the introduction
- ii) Comparison analysis of the FA profiles and LHI value bleached oils and crude oils presented in percentages and ratio in this paragraph would be better depicted in a table/figure. Table 1 and A1-6 are not enough as reference.
The fatty acid profile of the oils is shown in Table 2. We have decided to present only the results of the statistical analyses and discuss what changes have occurred. A summary of the raw results in a table or a graph will not add significant information to the article.
iii) or each sentence that refers to a table, make sure to put the table referred to in the appropriate place to make it easier for the reviewer. Please elaborate more and tabulated the actual results in the discussion section.
In our opinion, discussing individual fatty acid profile results is not the subject of the study. A broader discussion of the results with reference to their values is not in the area of the stated purpose of the study. It may cause difficulties in understanding the content provided.
- iv) Line 128 -misspelling ‘bleaching’
Corrected
- v) Line 136 – 137 – “This is interesting … previous experiment.” Please elaborate more.
The sentence was further clarified.
- vi) For Figure 1 and Figure 2 – Please include the loading plot to discuss further on the source of variation.
The drawings have been updated
vii) Line 217 – adjust figure caption to be in the same page as the figure.
This will be corrected at the graphic editing stage by MDPI.
Methods
- i) Line 237 -Table 2 should be in Results.
This is the characteristics of the material used in the study. Changes in the fatty acid profile due to bleaching were studied. This table must remain in the materials and methods section.
- ii) Line 244 – pH 3.2
Corrected
iii) Line 246 – Bentonite is not a proper noun
Corrected
- iv) Please include and explain the statistical analysis in the method section. Eg. MANOVA, PCA.
Added description to methodology section.
- v) Please put a flowchart instead of Figure 5 and Figure 6
The purpose of posting these photos was to show the consistency of the oils during the bleaching process. The insertion of a flowchart is not justified because these steps are not complicated and are described in the text.
Round 2
Reviewer 1 Report
The paper have been improved and revised according to the comments.
It can be accepted.
Author Response
Thanks again for your very valuable comments. Undoubtedly, their inclusion has increased the value of the publication considerably.
Reviewer 3 Report
1. Writers need to distinguish between nouns and proper nouns (line 55, 57, 60-61, 64-65, etc....)
2. ..."more than 75%". Give the reference
3. I still feel that the discussion needs to be done more carefully and precisely to facilitate the reader between lines 126-150.
Author Response
- Writers need to distinguish between nouns and proper nouns (line 55, 57, 60-61, 64-65, etc....)
We were not able to make grammar changes. We sent the article to language proofreading and made some significant grammatical changes.
2. ..."more than 75%". Give the reference
This information came from the publication [10] "Fatty acid composition of edible oils and fats". It was not given straightforwardly, only from the interpretation of that article.
3. I still feel that the discussion needs to be done more carefully and precisely to facilitate the reader between lines 126-150.
After reviewing the discussion again, we found a summary of the statistical description was missing. We supplemented the results section with a summary.
Thank you for your insightful review and comments, which helped us to remove errors from the publication.